# A Swedish Familial Genome-Wide Haplotype Analysis Identified Five Novel Breast Cancer Susceptibility Loci on 9p24.3, 11q22.3, 15q11.2, 16q24.1 and Xq21.31

**DOI:** 10.3390/ijms24054468

**Published:** 2023-02-24

**Authors:** Elin Barnekow, Johan Hasslow, Wen Liu, Patrick Bryant, Jessada Thutkawkorapin, Camilla Wendt, Kamila Czene, Per Hall, Sara Margolin, Annika Lindblom

**Affiliations:** 1Department of Clinical Science and Education, Karolinska Institutet, 11883 Stockholm, Sweden; 2Department of Oncology, Södersjukhuset, 11883 Stockholm, Sweden; 3Department of Molecular Medicine and Surgery, Karolinska Institutet, 17176 Stockholm, Sweden; 4Department of Neuroscience, Uppsala University, 75237 Uppsala, Sweden; 5Department of Biochemistry and Biophysics, Stockholm University, 17165 Stockholm, Sweden; 6Science for Life Laboratory, 17165 Stockholm, Sweden; 7Department of Medical Epidemiology and Biostatistics, Karolinska Institutet, 17165 Stockholm, Sweden; 8Department of Clinical Genetics, Karolinska University Hospital, 17164 Stockholm, Sweden

**Keywords:** GWAS, breast cancer, haplotype, familial, risk loci, *SMARCA2*, *GRIA4*, *TGIF2LX*

## Abstract

Most breast cancer heritability is unexplained. We hypothesized that analysis of unrelated familial cases in a GWAS context could enable the identification of novel susceptibility loci. In order to examine the association of a haplotype with breast cancer risk, we performed a genome-wide haplotype association study using a sliding window analysis of window sizes 1–25 SNPs in 650 familial invasive breast cancer cases and 5021 controls. We identified five novel risk loci on 9p24.3 (OR 3.4; *p* 4.9 × 10^−11^), 11q22.3 (OR 2.4; *p* 5.2 × 10^−9^), 15q11.2 (OR 3.6; *p* 2.3 × 10^−8^), 16q24.1 (OR 3; *p* 3 × 10^−8^) and Xq21.31 (OR 3.3; *p* 1.7 × 10^−8^) and confirmed three well-known loci on 10q25.13, 11q13.3, and 16q12.1. In total, 1593 significant risk haplotypes and 39 risk SNPs were distributed on the eight loci. In comparison with unselected breast cancer cases from a previous study, the OR was increased in the familial analysis in all eight loci. Analyzing familial cancer cases and controls enabled the identification of novel breast cancer susceptibility loci.

## 1. Introduction

Breast cancer (BC) is the most common cancer among women worldwide, with an incidence of 2.3 million cases per year [1]. The etiology of BC is multifactorial, with both genetic and environmental factors involved. Epidemiological studies have shown BC to be approximately twice as common in first-degree relatives of BC patients compared to the general population [2]. According to a twin study, inherited risk factors were estimated to influence the risk in 27% of BC cases [3]. Known mutations in the high penetrant genes, *BRCA1* and *BRCA2*, account for approximately 25% of familial BC [4]. Mutations in other high and moderate susceptibility genes, such as *TP53*, *PTEN*, *PALB2*, *ATM*, and *CHEK2*, explain 10% of familial cases [5]. In addition, a considerable number of low-risk genetic susceptibility loci have been identified through genome-wide association studies (GWASs) and account for another 18% [6,7,8]. Thus, the majority of BC heritability is unexplained.

Most known low-risk loci have been identified through GWAS, based on SNP association on a trait. Haplotype association has not been widely applied. Homogeneous populations are preferable when examining a haplotype effect on a trait, and our previous studies on cancer susceptibility in the Swedish population have found support for this approach [9,10,11]. In a GWAS of 3555 twins affected with cancer and 15,581 unrelated twins without cancer, seven susceptibility loci associated with general cancer risk were identified. One protective and six risk loci were identified, the latter with OR from 1.35 to 1.9 [9]. The GWAS of unselected cancer cases was followed by tumor-specific GWASs. A colorectal cancer GWAS identified two borderline significant loci and confirmed known colorectal cancer low-risk loci. Interestingly, subgroup analysis of familial cases, in contrast to the unselected analysis, could identify one novel significant colorectal cancer locus on 1q43 (OR 4.0; *p* 3.2 × 10^−8^) [10]. Recently, our haplotype GWAS on 3200 Swedish unselected BC cases and 5021 controls identified four significant loci—three well-known, the top three hits in BCAC on 10, 11, and 16, plus one novel BC susceptibility locus on 8p21.2. In studies of complex disease, enrichment of genetic risk alleles is expected in familial cases. This could be supported by increased ORs in the subgroup analysis of the four novel loci reported in familial BC [11]. Hence, we hypothesized that familial haplotype analysis, based on only familial cases from an unselected cohort in the Swedish population, could be a powerful approach to discovering further novel BC susceptibility loci.

Thus, the aim of this study was to identify BC risk haplotypes in the Swedish population and to compare the findings with the previous BC GWAS consisting of unselected BC cases [11]. Therefore, a GWAS was set up to analyze the 650 familial BC cases and 5021 controls selected from the previous study using unselected cases and controls.

## 2. Results

We identified eight significant familial BC susceptibility loci on chromosomes 9, 10, 11 (two loci), 15, 16 (two loci) and X (Table 1). In total, 1593 significant risk haplotypes and 39 risk SNPs distributed on the eight loci were found (Appendix A). Three loci, 10q25.13, 11q13.3 and 16q12.1, were previously known [6,11]. Five loci, 9p24.3, 11q22.3, 15q11.2, 16q24.1 and Xq21.31, were novel (Table 1, Figure 1, Figure 2, Figure 3, Figure 4 and Figure 5, and Appendix A). The three known risk loci on 10q25.13, 11q13.3 and 16q12.1 were previously well-described [11]. For comparison, the haplotypes of significant loci in the present familial BC study and the previous unselected BC study are presented with corresponding OR and *p*-values (Table 2) [11]. ORs in the familial analysis compared to the unselected BC analysis were increased (Table 2).

### 2.1. Locus 9p24.3

We identified 28 significant haplotypes, but no significant SNP, in a novel familial BC risk locus, 9p24.3 (Table 1 and Appendix A). The haplotype with the lowest *p*-value was located in an intergenic region between the *DMRT2*-gene and *SMARCA2*-gene but in immediate proximity to the latter (Figure 1). The haplotypes of various lengths share a region of seven SNPs upstream of the gene coding region (Figure 1).

### 2.2. Locus 10q26.13

We identified 143 significant haplotypes and 23 SNPs on 10q26.13, which were described in the subgroup analysis of a previous study (Appendix A) [11].

### 2.3. Locus 11q13.3

We identified 20 significant haplotypes, but no significant SNP, on 11q13.3, which were described in the subgroup analysis of a previous study (Appendix A) [11].

### 2.4. Locus 11q22.3

We identified one significant haplotype and three borderline significant haplotypes, but no significant SNP, in a novel familial BC risk locus, 11q22.3 (Table 1 and Appendix A). The locus totally overlaps the *GRIA4* region (Figure 2). The haplotypes of various lengths encompass the gene coding area (Figure 2).

### 2.5. Locus 15q11.2

We identified 1 significant haplotype and 22 borderline significant haplotypes of various lengths, but no significant SNP, in a non-coding region on 15q11.2 (Table 1, Figure 3 and Appendix A). All haplotypes share a central region of four SNPs (Figure 3).

### 2.6. Locus 16q12.1

We identified 1394 significant haplotypes and 16 SNPs on 16q12.1, which were described in the subgroup analysis of a previous study (Appendix A) [11].

### 2.7. Locus 16q24.1

We identified 3 significant haplotypes and 85 borderline significant haplotypes, but no significant SNP, in a novel familial BC risk locus, 16q24.1 (Table 1, Figure 4 and Appendix A)). Neither the most significant haplotype nor the borderline significant haplotypes were located in a gene coding region (Figure 4). The haplotypes of various lengths share a central region of two SNPs (Figure 4).

### 2.8. Locus Xq21.31

We identified 3 significant and 21 borderline significant haplotypes, but no significant SNP, in a novel familial BC risk locus, Xq21.31 (Table 1, Figure 5 and Appendix A). The haplotypes of various lengths are partly located in the gene coding region of *TGIF2LX*, and all except one haplotype share a region of five SNPs outside the coding area (Figure 5).

## 3. Discussion

In the search for BC risk genes in the Swedish population, we performed a haplotype GWAS on 650 familial invasive BC cases and 5021 controls by reanalyzing genotype data from a previous BCAC GWAS. We identified five novel risk loci on 9p24.3, 11q22.3, 15q11.2, 16q24.1 and Xq21.31 and confirmed three well-known loci on 10q25.13, 11q13.3 and 16q12.1 [8]. In a previous study, ORs were higher in familial cases compared to unselected cases, suggesting a genetic enrichment. The findings in this study confirmed our hypothesis that a haplotype GWAS of familial BC cases could be used to identify new risk loci (Table 1 and Table 2).

The majority of previous BC GWASs within the BCAC collaboration, including our samples, was performed with unselected BC cases [6,7,8]. The strategy to select only unrelated familial cases with enriched genetic risk alleles enabled the detection of novel susceptibility loci. Our previous studies on cancer susceptibility in the Swedish population have found support for this familial haplotype GWAS approach in relatively small and well-defined study populations [9,10,11]. The reason for this can be discussed. A founder mutation is not possible to detect with SNP analysis, whereas the combined nearby SNP effect—a haplotype effect—can enable the detection of a founder haplotype with an underlying founder mutation. This is well illustrated in all five novel loci with *p*-values of individual SNPs included in the significant haplotypes where all were non-significant (*p* > 2.7 × 10^−4^) (Appendix A). Haplotype analysis is a well-established method in the search for founder mutations [12]. In contrast, most GWAS has been based on SNP association on a trait and haplotype association analysis has not been widely applied. In isolated ethnic groups, the founder haplotype effect has been retained—so-called founder mutations. A founder mutation is supposed to be on a founder haplotype. The spectrum of pathogenic variants in *BRCA1* and *BRCA2* varies widely between populations and illustrates the population-specific founder effects. There are known founder mutations in *BRCA1* and *BRCA2.* The most well-known founder mutations are identified in the Ashkenazi Jewish population—two founder mutations in *BRCA1* (185delAG and 5382insC) and one in *BRCA2* (6174delT). In addition, two Swedish founder mutations have been identified in *BRCA1*—the most common, *BRCA1* 3171ins5, from the western region of Sweden and the *BRCA1* ins6kbEx13 identified in individuals of Walloon ancestry [13,14]. Homogeneous populations are preferable in haplotype analysis, which was why only Swedish samples were included in this study. This means these results constitute Swedish founder haplotypes that could be difficult to verify in other populations since haplotypes vary between populations. Thus, haplotype analysis is not preferable in a GWAS with mixed global populations. However, separate population-specific haplotype analysis in other homogenous populations could hopefully enable identification of risk haplotypes in the same loci.

Interestingly, apart from the three loci on 10q25.13, 11q13.3, and 16q12.1, we could not detect any of the previously published BC risk loci from large GWASs in unselected populations, probably because of our relatively small sample size, and therefore reduced power to detect very low-risk loci [6,7,8]. The clinical relevance of these loci can also be discussed individually. However, the combined effect of multiple low to ultralow BC susceptibility SNPs has been more promising. The polygenic risk score, comprised of 313 SNPs (PRS_313_), stratified individuals into different categories of risk, for which it was suggested to individualize mammographic screening in a small proportion of the population [15]. Recently, the clinical relevance of polygenic risk scores has been questioned because of population stratification problems and low discrimination [16,17]. In this study, we identified five novel loci associated with breast cancer, individually conferring a higher risk than previously presented. The role of these loci in future clinical practice remains unclear, but they point out genetic regions with potential underlying founder variants of interest for future studies. Our methodology of including only selected cases with familial BC increased the power to detect a genetic effect. The five novel familial risk loci did not reach a significant level in the unselected analysis because of the dilution of genetic risk factors by including sporadic cases, and therefore the sample size was too small for detection (Table 2). However, the OR was increased in the familial analysis compared to the unselected analysis, confirming a genetic association. The previously published risk loci on 8p21.2 did not reach significance in our analysis of familial cases, but as expected, the OR was increased (Table 2 and Appendix A) [11].

Identification of a significant haplotype in a haplotype BC GWAS indicates one or several genetic alterations in the region between the first and the last SNP, which confers an increased risk of BC. We suggest that the distribution of significant and borderline significant haplotypes in each locus can provide an indication of whether one or several underlying mutations are involved. The haplotypes on 16q24.1 share two centrally positioned SNPs, which could indicate one underlying genetic variant (Figure 4). In contrast, the loci on 11q22.3 share a large region of SNPs, which represent over 250 kB, which is why several variants could be involved (Figure 2).

Three of the novel loci were located in gene-coding regions and have previously been reported in relation to cancer, although the association with BC is not fully understood. The locus on 11q22.3 is centrally located in the *GRIA4-*gene region (Figure 2). *GRIA4* is involved in synaptic neurotransmission important for various neurological functions [18]. Variants in the *GRIA4* gene have been associated with developmental disorders and different forms of epilepsy [19]. Dysregulated signaling of *GRIA4* may play a role in cancer development by affecting different signaling pathways involved in proliferation and growth [20]. To our knowledge, the role of *GRIA4* in BC is yet unclear. However, its association with other cancer forms has been discussed. In colorectal cancer, a *GRIA4* promotor region is hypermethylated to a greater extent compared to colon adenoma [21]. In addition, hypermethylation of this promotor was associated with increased survival in HPV-related oropharyngeal squamous cell carcinomas [22]. The locus on 9p24.3 is partly located in the region of the *SMARCA2* gene, also named *BRM* (Figure 1). *SMARCA2* is a central component of the SWI/SNF complex that regulates the transcription of certain genes by chromatin remodeling [23]. The SWI/SNF complex is involved in numerous biological processes, including cell differentiation and proliferation [24]. The role of *SMARCA2* in BC is not conclusively elucidated but an oncogenic role has been suggested [25]. Overexpression of *SMARCA2* has been shown in BC tumors compared to normal breast tissue and knockdown of the gene in triple-negative BC cell lines reduced tumor formation and growth, supporting a tumor-promoting function [26]. In the locus Xq21.31, all the significant and borderline significant haplotypes are partly located on the *TGIF2LX* gene (Figure 5). *TGIF2LX* (transforming growth factor beta-induced factor 2 like, X-linked) belongs to the superfamily of homeodomain proteins that is involved in the regulation of several biological processes, including embryonic development, proliferation, and differentiation [27]. Overexpression of the gene has been shown to inhibit proliferation and angiogenesis in colon adenocarcinoma, thus suggesting that *TGIF2LX* could have a tumor suppressor role [28]. The risk locus on 15q11.2 and 16q24.1 is not located in a gene coding region. Sequencing of familial BC patients in further studies could detect the actual risk causal variant(s) within the candidate haplotypes. Because of the central haplotype position on the *GRIA4*-gene, the risk locus 11q22.3 is of particular interest.

In conclusion, our haplotype GWAS analysis in unrelated familial Swedish BC cases identified five novel BC risk loci and confirmed three well-known BC risk loci with higher ORs than previously presented. Our findings suggest that the haplotype GWAS methodology used in the setting of unrelated familial cancer cases and controls enabled the identification of novel BC susceptibility loci, even in a small study population. Nevertheless, haplotypes are not feasible for screening, but future studies in the haplotype regions could identify causative variants to be used in future genetic testing and risk prediction.

## 4. Materials and Methods

### 4.1. Study Population

Genotype data from a previous GWAS in three Swedish BC cohorts, KARMA, KARBAC1 and KARBAC2, were reanalyzed in this study. We included 652 unrelated familial BC cases (KARMA *n* = 516; KARBAC1 *n* = 66, KARBAC2 *n* = 70) and 5032 healthy controls from KARMA without family history of BC. The cohorts were thoroughly described elsewhere and previously analyzed in several BCAC studies [6,7,8,15,29,30,31,32,33]. Briefly, KARMA is a population-based cohort of 2712 cases and 5032 controls, recruited from October 2010 to March 2013 while performing a screening or clinical mammogram [31]. KARBAC1 is a hospital cohort of 394 consecutive BC cases recruited from October 1998 to May 2000 [32]. KARBAC2 is a cohort of 109 *BRCA1*- and *BRCA2-*negative BC cases recruited from a clinical genetic counseling department from February 2000 to January 2012 [34]. Familial BC was defined as an individual with invasive BC who self-reported one or more first-degree relatives with BC. The identical familial cases and controls were previously used for a subgroup analysis in a BC GWAS comparing frequencies and ORs of four identified susceptibility loci on chromosomes 8, 10, 11, and 16 in familial and unselected BC [11]. In addition, significant haplotypes identified in the present study were reanalyzed in the previous haplotype BC GWAS with 3215 unselected BC cases (familial and sporadic cases) from KARMA and KARBAC and the same controls as in the current study [11]. The studies were approved by the local ethical board and all individuals gave written informed consent (KARMA: Approved by the Ethical Committee of the Karolinska Institute, Dnr 2010/958-31/1; KARBAC1: Approved by the Ethical Committee of the Karolinska Institute, Dnr 98-232; KARBAC2: Dnr 2011/1686-32, and 2012/1453-32).

### 4.2. Genotyping and Quality Control

The individuals were genotyped using the Illumina Infinium OncoArray-500K B Bead-Chip [35]. The three cohorts shared 474,706 SNPs and the datasets were merged using PLINK v1.9 [36]. The genotypes were called using Illuminas’s “TOP” strand designation. In the quality control (QC) process 2332 variants were excluded using a <98% call rate threshold, 138,834 SNPs were excluded because of minor allele frequency (*p* < 0.01), and 634 markers were excluded because of deviation from Hardy–Weinberg equilibrium (*p* < 0.001). No individual was excluded because of missing genotype data. To identify ethnic outliers a multi-dimensional scaling (MDS) analysis for the four dimensions—coordinate1 (C1), C2, C3 or C4—was performed. Samples above +0.04 and below −0.04 were considered as ethnic outliers and were excluded from the material (2 familial BC cases and 11 controls). The final dataset comprised 332,906 SNPs, 650 familial BC cases, and 5021 controls. For SNPs and chromosomal positions, the reference panel GRCH37 was applied.

### 4.3. Statistics

We conducted a sliding window haplotype GWAS from window size 1 to 25 SNPs. Associations between the exposure (haplotypes of various lengths) and the outcome (familial BC) were examined using a logistic regression model in PLINK v1.07 [36]. The odds ratio (OR) was calculated to assess the effect of exposure on the outcome, and as the purpose was to detect risk loci, only OR > 1 was reported. Adjustment for population stratification was performed using principal coordinates C1, C2, C3, and C4 from the MDS analysis (see QC procedure) as covariates in the logistic regression model. The genome-wide significance, i.e., *p*-value < 5 × 10^−8^, was considered statistically significant [37]. A *p*-value between 9.9 × 10^−6^ and 5 × 10^−8^ was defined as borderline significant. No further correction for multiple testing was performed as the haplotypes of various lengths in each sublocus reflected the same genetic risk locus. A custom script was used to visualize the haplotype position (https://github.com/patrickbryant1/CMM/blob/master/hap_vis.py) (accessed on 15 November 2022) (Figure 1, Figure 2, Figure 3, Figure 4 and Figure 5).

## Figures and Tables

**Figure 1 ijms-24-04468-f001:**
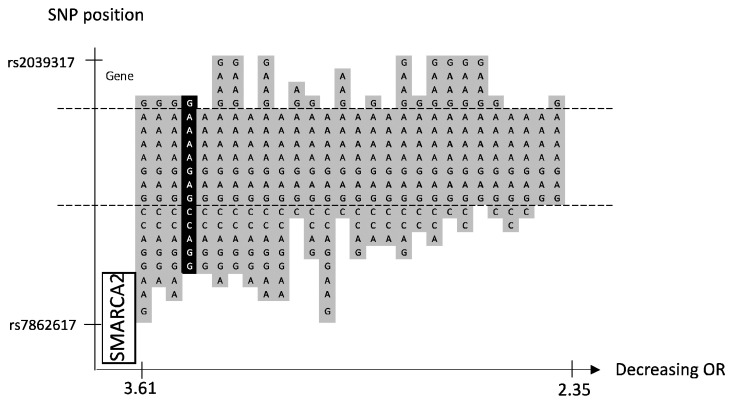
Chromosome 9, 28 significant haplotypes sorted by OR. The haplotype with the lowest *p*-value is marked in black and other significant haplotypes in gray. SNP position is on the *y*-axis. Gene position for SMARCA2 is indicated along the *y*-axis. The corresponding OR is on the *x*-axis. Because of numerous significant haplotypes in this locus, no borderline significant variants are shown. The dashed lines indicate the proximal and distal SNP that are shared by all significant haplotypes.

**Figure 2 ijms-24-04468-f002:**
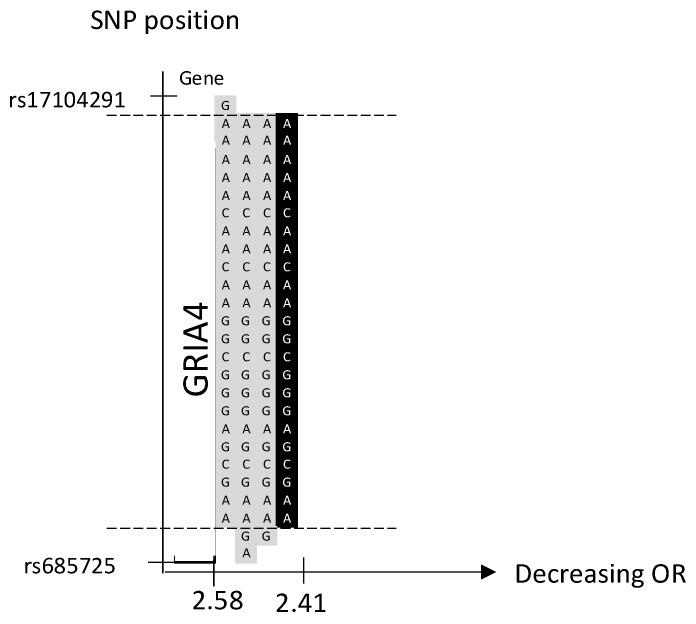
Chromosome 11, one significant haplotype and three borderline significant haplotypes sorted by OR. The significant haplotype is marked in black and borderline significant in light gray. SNP position is on the *y*-axis. Gene position for *GRIA4* is indicated along the *y*-axis. The corresponding OR is on the *x*-axis. The dashed lines indicate the proximal and distal SNP that are shared by the significant and borderline significant haplotypes.

**Figure 3 ijms-24-04468-f003:**
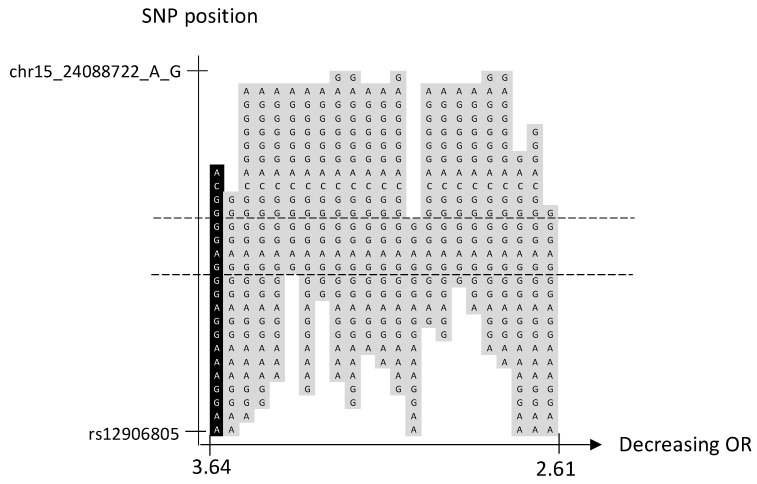
Chromosome 15, one significant and 22 borderline significant haplotypes sorted by OR. The significant haplotype is marked in black and borderline significant in light gray. SNP position is on the *y*-axis. The corresponding OR is on the *x*-axis. The dashed lines indicate the proximal and distal SNP that are shared by the significant and borderline significant haplotypes.

**Figure 4 ijms-24-04468-f004:**
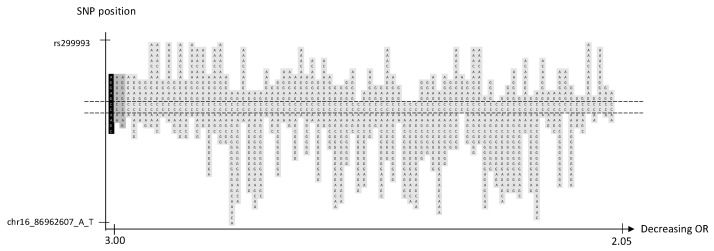
Chromosome 16, 3 significant and 85 borderline significant haplotypes sorted by OR. The haplotype with the lowest *p*-value is marked in black, other significant haplotypes in dark gray and borderline significant in light gray. SNP position is on the *y*-axis. The corresponding OR is on the *x*-axis. The dashed lines indicate the proximal and distal SNP that are shared by the significant and borderline significant haplotypes.

**Figure 5 ijms-24-04468-f005:**
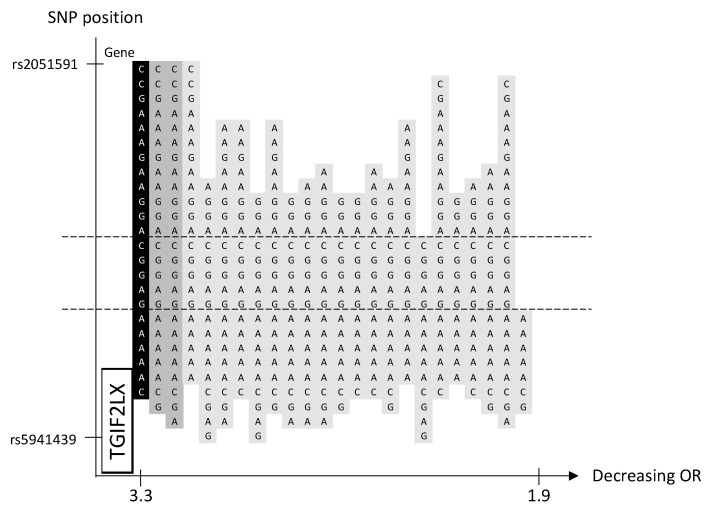
Chromosome X, 3 significant and 21 borderline significant haplotypes sorted by OR. The haplotype with the lowest *p*-value is marked in black, other significant haplotypes in dark gray and borderline significant in light gray. SNP position is on the *y*-axis. Gene position for *TGIF2LX* is indicated along the *y*-axis. The corresponding OR is on the *x*-axis. The dashed lines indicate the proximal and distal SNP that are shared by the significant and borderline significant haplotypes.

**Table 1 ijms-24-04468-t001:** Haplotypes with lowest *p*-value in eight familial breast cancer risk loci.

Locus	Gene	Haplotype	SNP1-SNP2	Size (kb)	F	OR	*p*-Value
9P24.3	*SMARCA2*	GAAAAGAGCCAGG	rs10122055-rs10119055	99	0.02	3.4	4.9 × 10^−11^
10Q26.13 *	*FGFR2*	GGG	rs4237533-rs10736303	2	0.42	1.5	8.5 × 10^−13^
11Q22.3	*GRIA4*	AAAAACAACAAGGCGGGAGCGAA	rs1373925-rs7116745	266	0.03	2.4	5.2 × 10^−9^
11Q13.3 *		GAAAAGGGA	rs7940177-rs614367	11	0.11	1.7	2.4 × 10^−9^
15Q11.2		ACGGGGAGGGAGGAAAGGAA	rs8037816-rs12906805	65	0.01	3.6	2.3 × 10^−8^
16Q12.1 *	*TOX3*	AG	chr16_52560213_A_G-chr16_52562811_A_G	3	0.23	1.5	4.0 × 10^−10^
16Q24.1		AGGAGCCAAGC	rs4843437-rs2059287	57	0.01	3	3.0 × 10^−8^
XQ21.31	*TGIF2LX*	CCGAAAGAAGGACGGAGAAAAAC	rs2051591-rs5941428	909	0.01	3.3	1.7 × 10^−8^

Each genetic locus with the gene in the area (if any), first (SNP1) and last (SNP2) and corresponding frequency (F), odds ratio (OR), and *p*-value. Reference panel GRCH37 for SNPs. * Previously published locus [11].

**Table 2 ijms-24-04468-t002:** Comparison of breast cancer risk loci in present familial and a previous unselected breast cancer GWAS.

	Familial BC		Unselected BC	
Locus	OR	*p*-Value	OR	*p*-Value
8p21.2 *	2.3	7.0 × 10^−5^	2.1	3.9 × 10^−8^
9p24.3	3.4	4.9 × 10^−11^	1.6	9 × 10^−4^
10q26.13 *	1.5	8.5 × 10^−13^	1.4	8.8 × 10^−21^
11q22.3	2.4	5.2 × 10^−9^	1.5	1.8 × 10^−5^
11q13.3 *	1.7	2.4 × 10^−9^	1.4	6.4 × 10^−13^
15q11.2	3.6	2.3 × 10^−8^	1.8	5.7 × 10^−4^
16q12.1 *	1.5	4.0 × 10^−10^	1.4	1.7 × 10^−18^
16q24.1	3	3.0 × 10^−8^	1.4	0.0094
Xq21.31	3.3	1.7 × 10^−8^	1.7	5.8 × 10^−4^

Risk loci significant in either familial or unselected breast cancer (BC) with the corresponding odds ratio (OR) and *p*-value. * Previously published locus [11].

## Data Availability

Access to the data is controlled. Variants that fulfilled our selection criteria can be found in the Appendix A. However, Swedish laws and regulations prohibit the release of individual and personally identifying data. Therefore, the whole dataset cannot be made publicly available. The data that support the findings of this study are available from the corresponding authors upon a reasonable request.

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
