# Peer review of "A Swedish Familial Genome-Wide Haplotype Analysis Identified Five Novel Breast Cancer Susceptibility Loci on 9p24.3, 11q22.3, 15q11.2, 16q24.1 and Xq21.31"

_ijms, 2023, doi:10.3390/ijms24054468_

Round 1

Reviewer 1 Report

The authors have downloaded the publicly available datasets and reanalyzed them with PLINK followed by statistical analysis. It would have been appreciated more if authors have come up with a novel method/datasets to address the locus associated with familial breast cancer.

The work submitted presents very little or no novelty. It may not have clinical relevance as well. Therefore it might not be of great interest to scientific community.

Author Response

Dear Reviewer,

Please see the attached file with response to your comments.

Reviewer 2 Report

Dear authors,

Your manuscript, "A Swedish Familial Genome-Wide Haplotype Analysis Identified Five Novel Breast Cancer Susceptibility Locus on 9p24.3, 11q22.3, 15q11.2, 16q24.1 and Xq21.31", aggroups results of previously published GWAS experiments for evaluating potential familial haplotypes related to breast cancer. I understood that a family approach requires a different perspective than an association study, which is the main contribution of this manuscript. So, I believe that some details need to be added as follows:

Major comments

1. Please add information about the statistical power of your sample. Over 600 families, how representative is it of the Swedish population?

2. Thinking of a clinical contribution, we need to apply novel findings in affordable techniques. Also, the theory of haplotypes suggests focusing on short fragments. Could we select haplotypes with a size of less than 5Kb and focus on their discussion? Do you have a plan to run a technical validation of some of these haplotypes? For example, sequence some samples using the Sanger method. I suggest discussing features of found haplotypes to propose some of them for eventual validation.

Minor comments

3. There are two Table 1. Please verify them.

4. The text "chromosome 9" was duplicated in the Figure 1 legend.

Author Response

Dear reviewer,

Please see the attached file with response to your comments.
